# The Usefulness of Wearable Sensors for Detecting Freezing of Gait in Parkinson’s Disease: A Systematic Review

**DOI:** 10.3390/s25165101

**Published:** 2025-08-16

**Authors:** Matic Gregorčič, Dejan Georgiev

**Affiliations:** 1Department of Neurology, University Medical Centre Ljubljana, Zaloška cesta 7a, 1000 Ljubljana, Slovenia; matic.gregorcic99@gmail.com; 2Division of Neurology, Medical Faculty, University of Ljubljana, Vrazov trg 2, 1000 Ljubljana, Slovenia; 3Artifical Intelligence Lab, Faculty of Computer and Information Sciences, University of Ljubljana, Večna pot 113, 1000 Ljubljana, Slovenia

**Keywords:** Parkinson’s disease, wearable sensors, freezing of gait

## Abstract

**Background:** Freezing of gait (FoG) is one of the most debilitating motor symptoms in Parkinson’s disease (PD). It often leads to falls and reduces quality of life due to the risk of injury and loss of independence. Several types of wearable sensors have emerged as promising tools for the detection of FoG in clinical and real-life settings. **Objective:** The main objective of this systematic review was to critically evaluate the current usability of wearable sensor technologies for FoG detection in PD patients. The focus of the study is on sensor types, sensor combinations, placement on the body and the applications of such detection systems in a naturalistic environment. **Methods:** PubMed, IEEE Explore and ACM digital library were searched using a search string of Boolean operators that yielded 328 results, which were screened by title and abstract. After the screening process, 43 articles were included in the review. In addition to the year of publication, authorship and demographic data, sensor types and combinations, sensor locations, ON/OFF medication states of patients, gait tasks, performance metrics and algorithms used to process the data were extracted and analyzed. **Results:** The number of patients in the reviewed studies ranged from a single PD patient to 205 PD patients, and just over 65% of studies have solely focused on FoG + PD patients. The accelerometer was identified as the most frequently utilized wearable sensor, appearing in more than 90% of studies, often in combination with gyroscopes (25.5%) or gyroscopes and magnetometers (20.9%). The best overall sensor configuration reported was the accelerometer and gyroscope setup, achieving nearly 100% sensitivity and specificity for FoG detection. The most common sensor placement sites on the body were the waist, ankles, shanks and feet, but the current literature lacks the overall standardization of optimum sensor locations. Real-life context for FoG detection was the focus of only nine studies that reported promising results but much less consistent performance due to increased signal noise and unexpected patient activity. **Conclusions:** Current accelerometer-based FoG detection systems along with adaptive machine learning algorithms can reliably and consistently detect FoG in PD patients in controlled laboratory environments. The transition of detection systems towards a natural environment, however, remains a challenge to be explored. The development of standardized sensor placement guidelines along with robust and adaptive FoG detection systems that can maintain accuracy in a real-life environment would significantly improve the usefulness of these systems.

## 1. Introduction

Parkinson’s disease (PD) is the second most prevalent neurodegenerative disease worldwide, affecting more than 6 million individuals and occurring in 1–2 individuals per 1000 people at any given time. Over the past 30 years, the prevalence of PD has increased by more than 2.5 times, which makes PD one of the major contributors to neurological disability today [1,2]. The main pathophysiological feature of PD is a progressive degeneration of dopamine-producing neurons in the substantia nigra pars compacta (SNpc) [3].

Clinical diagnosis of PD is based on the presence of bradykinesia, rest tremor and rigidity, in addition to changes in posture and gait [1]. In advanced stages of PD, treatment-resistant motor features like postural instability, freezing of gait (FoG) and falls become more prominent [3]. Gait disturbances and axial motor symptoms contribute to lower health-related quality of life (HRQoL scores) due to fear of injury associated with FoG and falls and disabling loss of mobility and independence [4,5]. FoG is one of the most debilitating motor symptoms in PD [6].

FoG is characterized by a temporary and brief interruption or reduction in forward motion of the feet despite the patient’s intention to walk [6] and most commonly occurs during gait initiation, turning, or when navigating narrow spaces. A systematic review of 128 studies published in 2023 identified turning as the most frequent trigger (28%), followed by doorway passing (14%) and dual-tasking while walking (10%). These manifestations significantly decrease mobility, increase fall risk and are correlated with a marked decline in quality of life in individuals experiencing FoG [7].

FoG can be seen in several neurological disorders but is most common in PD and atypical forms of parkinsonism, such as vascular parkinsonism, progressive supranuclear palsy, multiple system atrophy and corticobasal degeneration [6,8]. The presumed pathophysiology of FoG in PD involves dysfunction across multiple neural pathways that include pontomedullary reticular formation, the mesencephalic locomotor region, basal ganglia, cerebellum and cerebral cortex [6,9]. FoG is ultimately caused by sudden disruptions in the regulation of GABAergic inhibition in the basal ganglia, which affects activity in brainstem and locomotor centres [6].

The treatment of gait disturbances, such as FoG and falls, is often pharmacological but can also include physiotherapy, deep brain stimulation and cueing devices. To assess the effectiveness of these treatment options and the severity of gait disorders, reliable objective measurement methods are required [10]. An objective quantitative gait analysis system would improve currently used semiquantitative methods, which could refine and enhance diagnostics, therapy and prevention of FoG and other gait disturbances in PD [11]. To mitigate drawbacks of subjective rating methods, wearable technologies combined with machine learning algorithms are being developed as objective rating tools. These wearable sensors equipped with accelerometers, gyroscopes and magnetometers can continuously monitor patient movements in a home or laboratory setting and can therefore provide comprehensive real-time data, which can be used to accurately assess PD severity and improve treatment options [10,12].

Inertial measurement units (IMUs) are used in gait analysis and FoG detection due to their portability, low power consumption, low cost and ability to provide real-time kinematic data. These sensors are typically attached to segments of the lower limbs such as the thighs, shanks and feet, as well as the waist and wrists. IMUs can capture linear acceleration and angular velocity, which are then processed to determine joint angles and detect gait patterns. A study published in 2017 by Glowinski et al. [13] demonstrated that using IMUs with wavelet-based signal processing enables the extraction of gait features such as step phases, joint symmetry and transitions between stance and swing. These capabilities are critical in PD, where subtle gait changes precede and accompany FoG episodes. Recent research has applied IMU-based systems to detect FoG events by analyzing gait patterns in real time. To provoke FoG in experimental conditions, studies often utilize known trigger events, such as turning, gait initiation, or passing through narrow spaces, to identify specific FoG kinematic gait patterns [14,15,16,17,18,19,20,21,22,23,24,25].

In addition to FoG detection, IMUs can also be used to predict and quantify FoG. Different features can be extracted from IMUS, such as frequency components, entropy measurements and spatio-temporal parameters, which can then be processed by machine learning algorithms to determine normal gait, pre-FoG segments and actual freezing events. FoG prediction is based on the identification of subtle changes in gait parameters that occur seconds before freezing begins. The combination of sensor data and expert-labelled video recordings can then be used to quantify FoG by measuring the timing, frequency and duration of FoG episodes. This enables objective monitoring of FoG severity and frequency and therefore facilitates real-time interventions [26,27]. Despite the emerging use and development of wearable sensors for detecting and predicting FoG in PD, there is little consensus on the optimal sensor types, body location placement and data processing algorithms, which can range from machine learning to threshold approaches [10,12]. In addition, the complexity and diversity of FoG and technologies used to detect it further complicate comparisons and therefore the means to determine the most effective method to reliably detect FoG [12]. Many clinical studies have demonstrated promising results in controlled clinical trials, but the effectiveness and reliability of these systems in naturalistic contexts remains to be explored. The aim of this literature review is to systematically evaluate the effectiveness and limitations of current wearable technologies for detecting FoG in PD, identify the types of wearable technologies most commonly used to detect FoG in PD and examine the data processing methods used for FoG detection. In addition, we will investigate the current potential of real-life, naturalistic applications of wearable technologies for FoG detection in PD.

## 2. Review Methodology

A systematic literature review was performed according to the guidelines of the PRISMA statement. A database search of article titles and abstracts was performed by searching PubMed, IEEE Explore and ACM digital library. The final search was completed on the 26th of May 2025 for PubMed, the 31st of July 2025 for IEEE Explore and the 4th of August 2025 for ACM. Both medical and engineering approaches were allowed in the database screening. The final search strings are shown in Table 1. The only studies considered in the review were English, full-text, peer-reviewed original research articles. Articles that met the inclusion criteria were reviewed in full.

Articles were screened according to the following inclusion criteria:Studies focusing on FoG detection in PD patients using wearable technology in clinical or real-life settings;Original peer-reviewed articles in English.Studies were excluded based on the following exclusion criteria:Studies that do not involve wearable technology;Studies that do not focus on FoG detection;Non-human studies;Studies that do not provide sufficient details about the study design and results;Conference or workshop articles.

## 3. Results

### 3.1. Article Selection Process

An initial search of the databases identified 328 hits eligible for inclusion in the systematic review. Duplicate articles (N = 11) were removed, so 317 articles were screened based on title and abstract. After the screening process, 246 records were excluded, and the remaining 71 were reviewed in full. After the full review of the remaining articles, 28 were excluded for various reasons and 43 were included in the final review (Table 2). A complete overview of the selection process is summarized in Figure 1.

### 3.2. Demographic Data and Testing Environment

In the 43 fully reviewed studies, the number of tested subjects ranged from 1 [43] to 205 PD patients, with a median of 17 (IQR 10–41) [41]. The age of the subjects included in this review ranged from 62.5 [14,39] to 74 years [31], with a median of 68.9 (IQR 66.5–70.0) (mean = 68.5 ± 6.1). The mean age was not reported in 9 studies, while the other 34 did report the mean age of their patients. The Hoehn–Yahr stage was reported in less than half of the studies (N = 19), and it spanned from 2 [31] to 3.1 [33] with a median of 2.69 (IQR 2.5–2.95) (mean 2.54 ± 0.51).

Most studies (N = 28) focused solely on PD patients with previously clinically confirmed FoG episodes, while others (N = 7) used a broader approach and included FoG+ and FoG- PD patients [14,29,34,36,46,47,54] (Figure 2). In these studies, participants were classified as freezers or non-freezers based on the New Freezing of Gait Questionnaire (NFoGQ) [57]. One study that included FoG+ and FoG- PD patients did not report the exact method of FoG status classification [36]. Additionally, some studies (N = 8) did not explicitly report the FoG status of their PD subjects [18,21,30,35,37,43,44,48].

Most studies (N = 33) involved experimental setup and recruited participants based on a previously established study protocol, while the remaining ten studies used available public or institutional datasets [15,23,26,32,33,36,40,41,49,52]. Most commonly used datasets were DAPHNET [26,36,40,52], CuPiD [33,49], FP7 REMPARK [32], DeFOG [41], tDCS [41] and Hantao [41]. One study was multicentric [23], and in another, a dataset from a previously completed study was reused [15].

The majority of reviewed studies (N = 33) reported the medication states (ON/OFF) of the participants; however, ten studies [21,40,43,44,45,50,52,54,55,56] did not explicitly report the medication state. Across the studies reporting on the medication state, there was noticeable variability in the testing conditions. A notable portion of studies (N = 15) conducted testing in both medication states [17,18,23,25,27,30,31,32,34,35,37,38,41,42,48]. A comparable subset of studies (N = 11) evaluated the participants in an ON state while the patients were on their regular dopaminergic medication [14,20,22,24,26,28,29,33,46,49,53]. The patients were tested in an OFF state only or in a transition towards an OFF state in a smaller subset of studies (N = 7) [15,16,19,36,39,47,51].

Most of the reviewed studies (N = 33) conducted testing in a controlled laboratory environment or simulated real-life environment. Only ten studies [24,29,30,32,34,35,37,41,42,46] have attempted at least part of their testing in a real-life, naturalistic setting.

Commercially available sensors were utilized in most of the reviewed literature (N = 34), while custom IMUs were employed in a much smaller subset of studies (N = 9). The most utilized commercial IMU was the Mobility Lab Opal^TM^, which was used in nine studies [14,19,23,34,39,40,44,46,47].

## 4. Sensor Types, Locations on the Body for Sensor Placement and Gait Tasks

In this section we will present the results on the sensor types and the combinations of sensors, the locations on the body used for sensor placement and the gait tasks used.

### 4.1. Sensor Types and Combinations of Sensors

Most of the studies used accelerometers with combinations of other sensor types (90.7%) (Table 3). The most common combination was an accelerometer and a gyroscope (25.5%), followed by an accelerometer with a gyroscope and a magnetometer (20.9%). Other more complex combinations of accelerometers and other sensors were used in nine studies [16,18,20,25,28,41,51,54,56], including plantar pressure sensors, force sensing resistors and electromyography (EMG). Three studies did not use accelerometers but instead used plantar pressure sensors [15,22,48]. One study did not use any IMUs but instead utilized an SC sensor in combination with ECG [49]. In addition to inertial measurement units, two studies [41,51] used electroencephalography (EEG), while another study [47] used fNIRS (functional near-infrared spectroscopy) to monitor cortical activity during episodes of FoG.

### 4.2. Locations on the Body for Sensor Placement

The total number of articles in Figure 3 exceeds the total sum of reviewed articles (N = 43) as many studies (N = 23) used multiple body locations to place the sensors. The most commonly used body locations for sensors were the ankles (N = 14), the waist (N = 13), the feet (N = 13) and the shanks (N = 13). In the studies that used multiple body locations, the most common combinations were the waist together with ankle sensors [30,36,37,38], a combination of waist and shank sensors [19,28,47], a combination of shank, thigh and lower back sensors [26,52], and a combination of waist and feet sensors [28,47]. Only a smaller subset of studies (N = 7) used more complex approaches and included upper and lower body parts [30,34,37,38,45,47,51].

### 4.3. Gait Tasks

The gait task varied across the reviewed studies. In most of the studies, FoG-provoking elements were included in gait tasks, such as narrow corridors, obstacle navigation, sharp turns and start/stop movements. Some studies (N = 11) also included motor and cognitive dual-tasking in their gait testing [14,15,16,19,21,22,23,33,34,46,47]. The most commonly used standardized gait task was the TUG (Timed Up and Go Test) [58]. A considerable number of the reviewed articles (N = 17) conducted at least some of the motor testing without specific gait tasks but instead used a real or simulated real-life context to evaluate FoG [20,24,25,26,27,29,30,31,32,34,35,37,40,41,42,45,52].

### 4.4. Performance Metrics for Sensors and Sensor Combinations

In this section, we will present the best performance metrics for each sensor or sensor combination across the reviewed studies. Performance reporting was not standardized throughout the studies. In addition to the standard performance metrics (sensitivity, specificity and accuracy), other metrics such as the F1 score, AUC, error rate and others are grouped under “other performance metrics” (Table 4).

An accelerometer was the most used sensor type (N = 39), with ten studies [26,29,32,35,36,40,42,44,46,52] using only an accelerometer and 29 using accelerometers in combination with other sensor types. Studies that employed an accelerometer as the only sensor type reported specificity ranging from 67.0% [52] to 97.9% [40], sensitivity ranging from 81.6% [42] to 98.5% [40], and accuracy ranging from 81% [29] to 98.5 [40]. Three studies [15,22,48] did not use an accelerometer but instead used plantar pressure sensors, and one study used ECG with an SC sensor [49]. The most common sensor combination was an accelerometer and gyroscope (N = 11) [14,17,21,24,31,33,37,38,39,43,45], followed by a combination of an accelerometer, gyroscope and magnetometer (N = 9) [19,23,27,30,34,47,50,53,55]. Studies that employed multimodal sensor setup reported sensitivity ranging from 68.3% [14] to ~ 100% [21] and specificity ranging from 42% [19] to ~ 100% [21]. The remaining nine studies used unique combinations of an accelerometer and other sensor types, specific to each individual study.

## 5. Data Analysis Algorithms

Most studies (N = 31) used machine learning models to process the data from the FoG detection systems. These models involved supervised learning techniques that used features extracted from the wearable sensors. The most used machine learning models were the Convolutional Neural Network (CNN), which was used in ten studies [17,22,24,31,32,40,45,54,55,56], and the Support Vector Machine (SVM), which was used in nine studies [21,23,26,27,29,35,44,51,52]. Threshold-based algorithms, which use fixed rules to detect events when the signal exceeds or falls below a set value, were used in a smaller subset of studies (N = 9) [18,19,25,34,43,46,47,49,53]. The combination of machine learning and threshold approaches was employed in two studies [14,30], and one study used a proprietary FoG detection algorithm [42] and did not precisely explain the used algorithm.

## 6. FoG Detection in a Real-Life Naturalistic Environment

A naturalistic real-life environment was the focus of only nine studies [24,29,32,34,35,37,41,42,46]. The number of tested patients varied significantly from 10 [42] to 125 [41]. The ON/OFF medication state was not consistently controlled in one study [41], while four studies conducted testing in both states [32,35,37,42], and another four studies [24,29,34,46] tested the patients in an ON state only.

Out of the nine studies that focused on detecting FoG in a real-life naturalistic setting, three studies [29,32,35] used a single wearable sensor placed on the waist for testing. In contrast, the remaining six studies used multiple sensors placed on various body parts. Sensor placement in these studies is summarized in Figure 4.

The sum of studies that focused on real-life FoG detection in Figure 4 exceeds the total sum of articles (N = 9) because most articles (N = 6) used multiple body locations to place the sensors. The waist was the most common body location for sensors; three studies used a single wearable sensor in this location [29,32,35], and four studies used a combination of this location with other body sites [24,37,41,42].

Table 5 summarizes the best reported performance metrics and algorithms for data analysis for each sensor type or sensor type combination in real-life studies. Different performance metrics were used in different studies. Any metrics other than sensitivity, specificity and accuracy are referred to as “other performance metrics”. An accelerometer was identified as the most used stand-alone sensor type in five studies [29,32,35,42,46]. In four studies [24,34,37,41], the accelerometer was used in combination with a gyroscope or gyroscope and magnetometer. The specificity of the use of an accelerometer ranged from 79% [29] to 88.3% [32], and the sensitivity ranged from 73.1% [42] to 87.7% [32]. The other real-life studies that used a multimodal sensor approach reported a variety of different performance metrics (Table 5). Three naturalistic studies [29,32,35] employed the use of a single STAT-ON^TM^ wearable device, while others used research-grade sensors that were not embedded in a single wearable device.

## 7. Discussion

The main aim of this systematic review was to assess the current limitations and effectiveness of wearable sensors to detect FoG in PD patients. A total of 43 articles were fully reviewed to determine the best sensor type and their combinations, body placement locations, data analysis algorithms and current potential of real-life applications of wearable technologies for FoG detection in PD patients.

### 7.1. Sensor Types and Performance

A notable finding of this review is the predominant use of accelerometers as a fundamental component in most FoG detection systems. Among the 43 reviewed studies, over 90% used an accelerometer as a standalone wearable sensor or in combination with other sensor types, which corresponds to the findings of other studies [12]. The accelerometer measures the acceleration of objects along its reference axis [59]. In addition, it is cost-effective, non-invasive and widely available, which explains its frequent use in many studies. An accelerometer was also by far the most frequently used standalone sensor type (23.2%), followed by plantar pressure sensors [15,22,48]. However, most of the reviewed studies (69.7%) employed a multimodal sensor configuration approach to acquire more data and improve detection accuracy.

Studies that used an accelerometer as a standalone component in their FoG detection system reported a wide range of performance metrics. The specificity of the systems in detecting FoG varied from 67.0% [52] to 97.9% [40], sensitivity ranged from 81.6% [42] to 98.5% [40], and overall accuracy spanned from 81% [29] to 98.5% [40]. The best overall performance from the accelerometer-based sensor configuration was reported by Ashfaque et al. 2021, achieving 98.5% sensitivity, 97.9% specificity and 98.5% accuracy. In comparison, the best reported performance metrics from studies that used plantar pressure sensors as their main configuration were reported by Park et al., 2024, with 88% sensitivity, 99% specificity and 99% accuracy. These findings suggest that the type of sensor used in a single sensor configuration system does not substantially impact performance [12].

Among the studies that used multimodal sensor configurations, the most frequent combination was an accelerometer and a gyroscope, which was used in 11 studies (25.5%). The combination of an accelerometer, gyroscope and magnetometer was reported in nine studies (20.9%). Other, more complex configurations, such as combinations of accelerometers and gyroscopes with ECG, EEG, EMG, SC, force resisting sensors or fNRIS, were rarely used probably due to the complexity of using these systems.

The sensitivity of the accelerometer–gyroscope combination ranged from 68.3% [14] to ~100% [21], while specificity ranged from 67% [33] to ~100% [21]. In the studies using accelerometer–gyroscope–magnetometer systems, the sensitivity ranged from 80% [23] to 98% [19], and the specificity spanned from 42% [19] to 98% [30]. The highest overall performance among the multimodal sensor configurations was reported by Chomiak et al. 2019 with nearly 100% sensitivity and specificity and an average error rate of less than 5%. Note that Chomiak et al. 2019 used a single device (iPod touch) equipped with both an accelerometer and gyroscope.

Importantly, while multimodal sensor approaches may offer larger data acquisition and marginal improvements in detection accuracy regarding FoG and overall system performance, they also introduce complexity and potential patient discomfort, particularly in a more naturalistic environment, which could limit the scalability of such systems outside of a controlled laboratory environment [60].

### 7.2. Locations on the Body Where Sensors Were Placed

The waist, ankles, feet and shanks were the most frequently used locations where the sensors were placed. These findings likely reflect a balance between the data quality acquisition from these sites and patient comfort related to it. This trend is evident in studies that conducted at least part of their testing in a naturalistic real-life environment [29,32,34,35,37]. In contrast, studies that employed complex multimodal sensor configurations with sensors placed on upper and lower body parts presented greater variability in their sensor placements. This diversity underlines the lack of standardized guidelines for optimal sensor placement for FoG detection. The vast majority of studies relied on machine learning algorithms for data analysis, so inconsistency in sensor placement may restrict the comparability of their findings and contribute to inconsistent performance with similar data analysis methods [61]. Direct and systematic comparison of different sensor placements under the same experimental conditions could optimize system performance and patient comfort [61,62].

### 7.3. Gait Tasks

Structured laboratory-based gait tests designed to provoke FoG were predominantly used in most studies (N = 34). Furthermore, a notable subset of these studies (N = 11) also added additional cognitive tasks to maximize the probability of triggering FoG [14,15,16,19,21,22,23,33,34,46,47]. This approach reflects the objective of maximizing the performance of detection systems in controlled environments. Only nine naturalistic studies aimed to detect FoG episodes without specific supervised structured gait tasks, which improves their ecological validity but also presents challenges for detection system development due to unexpected patient behaviour and user compliance issues in unsupervised settings. The development of standardized gait tasks that also include common real-life gait elements could minimize the gap between system development and its generalization in real-life contexts. The incorporation of simulated real-life activities is already demonstrated in multiple reviewed studies [20,24,25,26,27,31,40,42,45,52] but lacks overall standardization, which would improve the comparability of findings between different detection systems.

### 7.4. Medication State

The medication states of PD patients should be an important consideration for FoG detection systems since FoG can be broadly categorized as either ON FoG that occurs in an ON medication state and OFF FoG that occurs in an OFF medication state, and it is more common [63]. Most of the reviewed studies (N = 33) reported the medication state. In 15 studies, testing was carried out in both medication conditions [17,18,23,25,27,30,31,32,34,35,37,38,41,42,48]. In 11 studies, testing was performed in the ON medication state only [14,20,22,24,26,28,29,33,46,49,53], and in 7 studies, testing was performed in the OFF medication state only [15,16,19,36,39,47,51]. The two FoG phenotypes may have different underlying mechanisms that are still poorly understood and could require different therapeutic and detection approaches [64], so accurate reporting of medication states should be adopted.

### 7.5. Patient FoG Status

Most reviewed studies (N = 35) reported the FoG status of their participants as FoG+ or FoG-. Among these studies, 28 focused only on patients that had previously clinically confirmed FoG episodes, likely reflecting the objective to capture as many FoG episodes as possible to maximize the detection performance of their systems. Seven studies that also included FoG- patients [14,29,34,46,47,54] classified their participants as freezers or non-freezers based on the New Freezing of Gait Questionnaire (NFoGQ) [57], except for one study that did not report the exact classification method used [36]. This classification may be arbitrary in this context since self-assessment of FoG may be unreliable due to recall bias and the possible cognitive decline of patients. Classifications of FoG status of PD patients should be based on robust and objective measurements [7,65]. One of the important factors limiting FoG research is the fact that FoG is a stochastic nature of this phenomenon that depends on many factors, such as environmental (e.g., the wideness of the space, light conditions and patterns on the floor), psychological condition (level of anxiety and depression, attention and novelty) and medication status [7,66]. Therefore, even though patients might report FoG status in their everyday lives, these episodes might not be easily triggered in a laboratory setting, making it difficult to explore in a controlled fashion.

### 7.6. Data Analysis Algorithms

The machine learning models used in most studies were fundamental data analysis methods (N = 31), showing growing interest in data-driven methods to increase the validity of FoG detection system performance. In contrast, threshold-based algorithms were used in a smaller subset of studies (N = 9) [18,19,25,34,43,46,47,49,53]. These methods are computationally effective and easier to deploy and implement into detection systems [12]. However, they are less reliable across variable populations because of high false positive rates with a lower threshold cutoff and the opposite with a higher threshold cutoff [12,67]. Threshold-based algorithms may still be valuable in real-life long-term monitoring for their simplistic and low-cost approach. This especially goes for the personalized threshold approach that has the potential to determine the best cut-off values based on patients’ own data [67].

It is evident that this research field is transitioning towards learning-based approaches with higher validity and adaptability. Direct comparisons of different machine learning models using the same data obtained from standardized experimental conditions could help to optimize data analysis methods and lead to better clinical adoption of these systems.

### 7.7. Potential Use of Sensors for FoG Detection in Real-Life Naturalistic Settings

The detection of FoG in naturalistic environments would be clinically valuable for long-term data collection, patient monitoring and potentially early gait deterioration detection [68], but this area of research seems to be particularly challenging. This review identified a small subset of studies (N = 9) [24,29,32,34,35,37,41,42,46] that conducted at least part of their testing in an unsimulated real-life context.

Most of the naturalistic studies (N = 6) employed multi-sensor configurations consisting of multiple accelerometers or combinations of accelerometers, gyroscopes and magnetometers. The remainder of the naturalistic studies (N = 3) utilized a single waist-worn accelerometer; notably all three of these studies used a STAT-ON^TM^ integrated sensor device [29,35,62]. Apart from the three studies that used a STAT-ON^TM^ device, all other naturalistic studies used research-grade sensors that were not integrated into a single smart device. In this context, future research on FoG detection in real-life settings should explore the potential use of sensors embedded into single smart devices (smartwatches and smartphones) for better patient comfort and long-term continuous monitoring.

The findings of the reviewed naturalistic studies support the viability of real-world monitoring and consistent sensor performance. However, when compared to performance in controlled laboratory environments with a similar sensor setup, it is evident that performance is less consistent and reliable. In the context of testing, real-life settings and controlled environments differ by signal noise, unexpected movements and user compliance, particularly in unsupervised scenarios [69].

Future research should prioritize the transition from controlled laboratory monitoring towards an unsupervised naturalistic context, which requires exploration and validation of robust, patient-friendly sensor systems embedded in smart devices and adaptive machine learning algorithms that can maintain accuracy in diverse real-life conditions.

### 7.8. Limitations

The main objective of this review was to evaluate the usefulness of wearable sensors for detecting FoG in patients with PD. FoG is a symptom that is not unique to PD but can also be observed in other neurological disorders, including other parkinsonisms. Therefore, a broader, comparative perspective on FoG in different neurological diseases could be the aim of a review that could shed light on the differences in FoG in different neurological diseases. The number of articles included in this review is rather small. However, we chose to investigate the utility of wearable sensors in detecting FoG in PD by applying strict inclusion and exclusion criteria for the studies included in this review. Furthermore, due to the heterogeneity of the studies, we chose to summarize the current state of knowledge in a systematic way rather than performing a meta-analysis, although we followed the PRISMA guidelines for systematic reviews.

## 8. Conclusions

This systematic review underlined the current progress and remaining challenges in the use of wearable sensors to detect FoG in PD patients. Among the 43 reviewed studies, accelerometers were found to be the fundamental component of most standalone and multimodal sensor configurations. Out of all reviewed sensor combinations, the best performance was reported by Ashfaque et al., 2021, who used an accelerometer as a single sensor type configuration, and Chomiak et al., 2019, who utilized a single iPod Touch equipped with an accelerometer and gyroscope. More complex multisensory configurations do not seem to cause significant performance gains and introduce potential concerns for long-term monitoring in naturalistic settings due to increased complexity and patient discomfort.

Sensor placement sites on the body lack standardized guidelines; however, the waist, ankles, feet and shanks seem to be the most frequent choices, particularly in studies that focused on real-world applications of their FoG detection systems. The lack of uniform guidelines for sensor placements on the body may cause increased variability in performance across similar machine learning algorithms.

Although nine studies reported promising performance metrics for real-world FoG detection systems, this area seems to remain technically challenging and underexplored. Studies that focused on FoG detection in a naturalistic context reported lower and more variable performance metrics, likely because of factors such as increased signal noise, user compliance and unexpected activities, which are otherwise not present in controlled laboratory environments.

Therefore, this systematic review identified multiple high-performing sensor configurations for FoG detection in PD patients in a controlled environment but also the challenge of transitioning these systems into a real-world environment for long-term and continuous gait monitoring. Future research in this area should focus on developing uniform sensor body placement guidelines, robust and patient-friendly sensor configurations and adaptive machine learning algorithms capable of maintaining accuracy in a naturalistic setting. Improvements in systems that are already performing well in laboratory environments would make real-life FoG detection in PD patients more viable and clinically impactful.

## Figures and Tables

**Figure 1 sensors-25-05101-f001:**
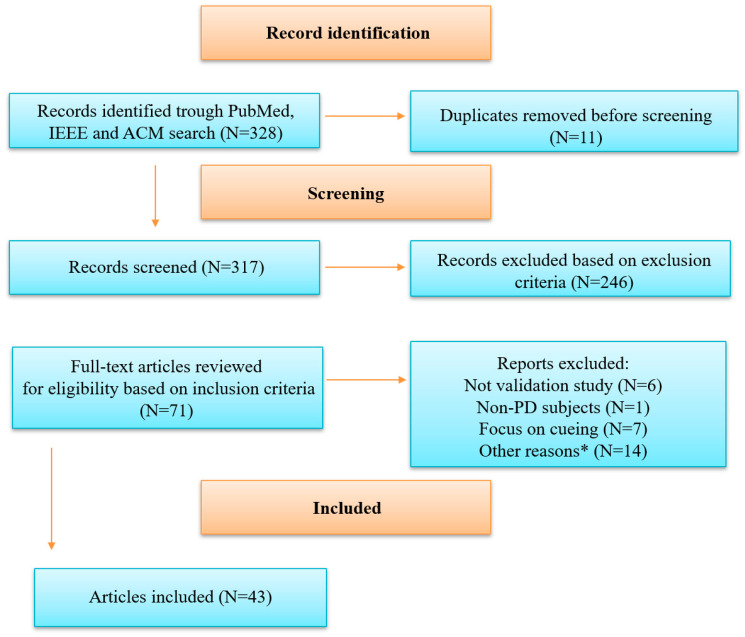
Diagram of study selection process.PD = Parkinson’s disease. * Other reasons included focusing on axial symptoms (N = 1), focusing on treatment interventions (N = 1), not focusing on wearable technology (N = 7) and full English text not available (N = 5).

**Figure 2 sensors-25-05101-f002:**
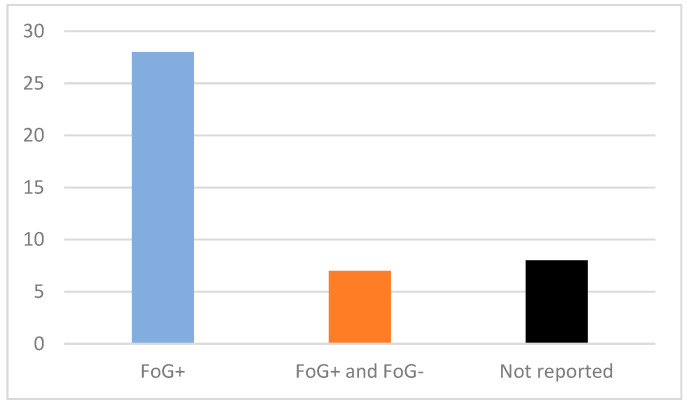
Article distribution based on subjects’ Freezing of Gait (FoG) status. Y-axis—Number of articles.

**Figure 3 sensors-25-05101-f003:**
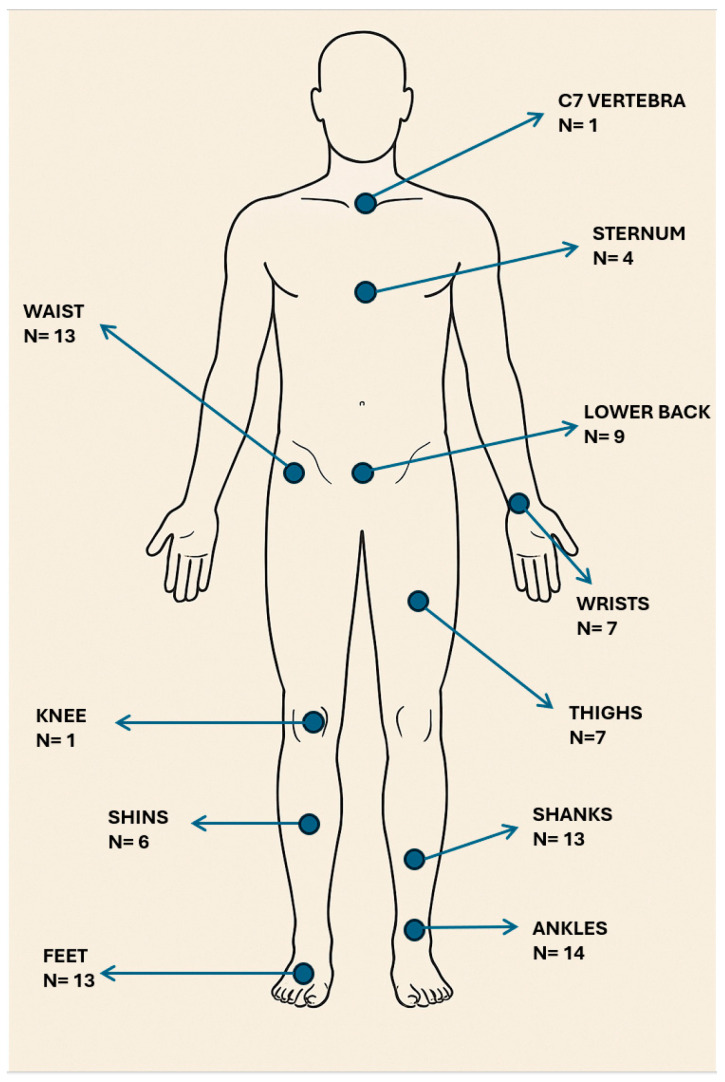
Anatomical position of the sensors in all the reviewed studies. The locations of sensors are marked with blue dots that relate to appropriate names of the body parts. The number below the name of the location represents the number of articles that applied wearable sensors to that body part.

**Figure 4 sensors-25-05101-f004:**
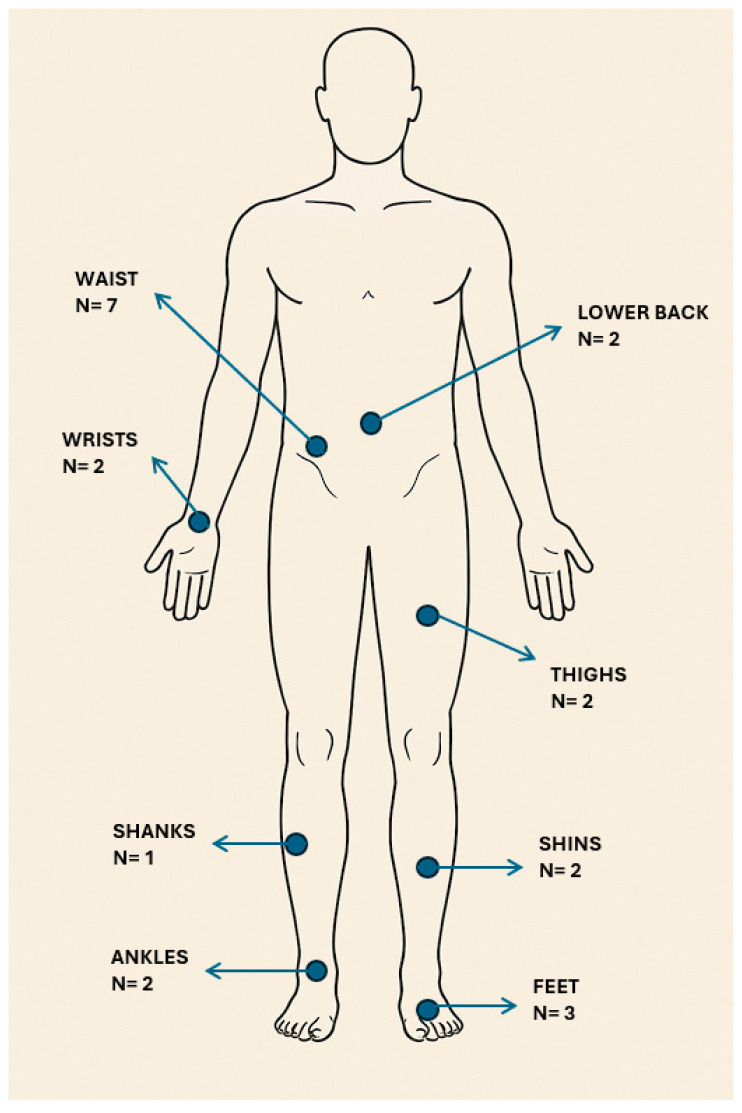
Anatomical positions of the sensors in the studies on the detection of FoG in a real-life, naturalistic setting. The locations of the sensors are marked with blue dots that relate to appropriate names of the body parts. The number below the name of the location represents the number of naturalistic studies that applied wearable sensors to that body part.

**Table 1 sensors-25-05101-t001:** Search string of Boolean operators used in PubMed, IEEE Explore and ACM digital library search.

Database	Search String	No. of Records
PubMed	((“freezing of gait”[Title/Abstract] OR “FoG”[Title/Abstract]) AND (“wearable sensor”[Title/Abstract] OR “wearable sensors”[Title/Abstract] OR “wearable device”[Title/Abstract] OR “wearable devices”[Title/Abstract]) AND (“Parkinson’s disease”[Title/Abstract] OR “Parkinson disease”[Title/Abstract]))	101
IEEE Explore	(“freezing of gait” OR “FoG”) AND (“wearable sensor” OR “wearable sensors” OR “wearable device” OR “wearable devices”) AND (“Parkinson’s disease” OR “Parkinson disease”)	91
ACM digital library	(“freezing of gait” OR “FoG”) AND (“wearable sensor” OR “wearable sensors” OR “wearable device” OR “wearable devices”) AND (“Parkinson’s disease” OR “Parkinson disease”)	136

**Table 2 sensors-25-05101-t002:** A summary of the reviewed articles. All the data are from original research articles.

Author, Year	Demographic and Clinical Data	Sensor Type and Model	Sensor Location	Test	Main Results	ON/ OFF	CLT/ RL	Algorithm
Ren et al. [28], 2022	12 PD-FoG+ Mean age: 66.75 Mean H&Y: 2.67	Accelerometers, gyroscopes, force sensing resistors Commercial BMX055	Waist, thighs, shanks, feet, insoles—1 sensor per body part	Random gait test, TUG	Sensitivity: 78.39% Specificity: 91.66% Accuracy: 88.09% Precision: 77.58% F-score: 77.98%	ON	CLT	ML: random forest
Zampogna et al. [29], 2024	71 PD, 33 PD-FoG+, 29 PD-FoG- Mean age: 69 Mean H&Y: 2	Accelerometer Commercial STAT-ON^TM^	Waist (left side)—1 sensor	Daily activities for 5–8 days	Sensitivity: 0.82 Specificity: 0.79 Accuracy: 0.81	ON	RL	ML: support vector machine
Pardoel et al. [15], 2024	21PD-FoG+ Dataset: Pardoel et al. 2022. Mean age: 72.4 Mean H&Y: NR	Plantar pressure insoles Commercial Tekscan	Both feet—1 sensor per foot	Walking on freeze-inducing path with cognitive dual-tasking	Sensitivity: 77.68%, Specificity: 79.99%, FOG identification: 86.84% Predicted FoG: 0.94s before onset	OFF	CLT	ML: RUS boost ensemble of decision trees
Pardoel et al. [16], 2021	11 PD-FoG+ Mean age: 72.7 Mean H&Y: NR	Accelerometer, gyroscope and plantar pressure sensor Commercial Tekscan and Shimmer3	Insoles in shoes—1 sensor per shoe Shanks—1 sensor per shank	FoG provoking walking path with dual-tasking, narrow passages, turns, stops and starts	Sensitivity: 76.4% Specificity: 86.2% FOG-only detection: 93.4%	OFF	CLT	ML: RUS-boosted decision tree ensemble
Koltermann et al. [17], 2023	11 PD-FoG Mean age: NR Mean H&Y: NR	Accelerometer and gyroscope Commercial Ultigesture IMU	Both ankles—2 sensors per ankle	Walking tests designed to trigger FoG	F1 score: +13.4% accuracy +10.7% FPR reduced by 85.8%	ON/ OFF	CLT	ML: multi-input convolutional neural network
Marcante et al. [18], 2020	20 PD Mean age: 68.6 FoG status: NR Mean H&Y: >3	Plantar pressure sensors and accelerometer Commercial Moticon GmbH	Both feet (in-shoe insoles with 13 sensors and 1 accelerometer)	TUG, 360° turn, 2MWT, door opening, drinking task, standing and walking under various conditions	Sensitivity: 96%; specificity: 94% FPR: 6% FNR: 4%	ON/ OFF	CLT	Threshold-based detection algorithm using force, COP, vertical acceleration signals
Antonini et al. [30], 2023	65 PD Mean age: 65.8 FoG status: NR Mean H&Y: NR	Accelerometer, gyroscope and magnetometer Commercial PD-Monitor®	Wrists, ankles, waist—1 sensor per body part	Phase I: supervised tasks in hospital Phase II: free-living activities for up to 3 days	Accuracy: 96% Specificity: 98% Sensitivity: 83%	ON/ OFF	CLT/ RL	ML: naive Bayes classifier, ROC-based thresholding
Delgado-Terán et al. [31], 2025	21 PD-FoG+; mean age: 74 Mean H&Y: 0.5	Accelerometer and gyroscope Custom IMUs	Right ankle—1 sensor	Walking, turning, sitting, standing, household tasks	AUROC: 0.89–0.96 (5Fold-CV), 0.90–0.93 (LOSO) Sensitivity: 96.5%; F1-score: 92.1%	ON/ OFF	Semi RL, VV	ML: convolutional neural network (machine learning)
Borzì et al. [32], 2023	21 PD-FoG+; mean age: 69.3 Mean H&Y: NR Dataset used: FP7 REMPARK (main test set)	Accelerometer Commercial IMUs in REMPARK Project	Waist—1 sensor	Free living activities	Main test set: 50% FoG predicted 3.1s before onset, 50% FoG detected with 0.8s delay Sensitivity: 0.877; specificity: 0.883	ON/ OFF	RL	ML: multi-head convolutional neural network
Palmerini et al. [33], 2017	11 PD-FoG+ Mean age: 67.7 Mean H&Y: 3.1 Dataset used: CuPiD	Accelerometer and gyroscope Commercial IMUs in CuPiD project	Left and right ankles; lower back—1 sensor per body part	Multiple walking conditions turns, dual tasks, narrow corridors	Mean AUC: 0.76, sensitivity: 0.83, specificity: 0.67	ON	CLT	ML: linear discriminant analysis
Krasovsky et al. [19], 2023	14 PD-FoG+ Mean age: 65.1 Mean H&Y: NR	Accelerometer, gyroscope and magnetometer Commercial Mobility Lab Opal^TM^	Waist and both shanks—1 sensor per body part	Walking tasks with turns, dual-tasking, figure-eight patterns, voluntary stops	Sensitivity: 98%, specificity: 42%, balanced accuracy: 70.2% SLT occurred ~1.8s before FoG onset	OFF	CLT, VV	Wavelet coherence analysis (threshold)
Slemenšek et al. [20], 2024	9 PD-FoG+; mean age: 67 Mean H&Y: 2.7	Accelerometers, gyroscope and muscle activity sensors Custom IMUs	Below both knees—multisensory strip per knee	10–15 min gait trials with walking, turning, door crossing	Sensitivity: 2.7% Specificity: 97.2% Accuracy: 95.0% F1-score: 0.023 Mean detection delay: 261 ms	ON	CLT	ML: NN+RNN+PS model
Chomiak et al. [21], 2019	21 PD, 9 HC Mean age: NR FoG status: NR Mean H&Y: NR	Gyroscope and accelerometer in iPod Touch Commercial device	Thigh—1 sensor in pocket	Walking or stepping in place with turning, dual-tasking and cup carrying	Model B: <5% mean error rate, 0% mode error rate, ~100% sensitivity and specificity	NR	CLT	ML: RQA + SVM with Monte Carlo cross-validation
Borzì et al. [27], 2021	11 PD-FoG+ patients Mean age: 73 Mean H&Y: 2.7	Accelerometer, gyroscope and magnetometer Custom IMUs	Both shins—2 sensors per body part	Timed Up and Go test in free-living-like setting with narrow corridor and door	Pre-FOG detection in LOSO; sensitivity: 84.1–85.5%, specificity: 85.9–86.3%, accuracy: 85.5–86.1%	ON/ OFF	Semi-CLT, sim. RL, VV	ML: wrapper feature selection + SVM and LDA classifiers
Mancini et al. [34], 2021	Study I: 45 PD (27 FoG+ and 18 FoG-) and 21 HC Mean age: PD 70.1 Study II: 48 PD (23 FoG+ and 25 FOG-) Mean age PD 68.6 Mean H&Y: 2–4	Accelerometer, gyroscope and magnetometer Commercial Mobility Lab Opal^TM^	Study I: feet, shins, wrists, sternum, lower back—1 sensor per body part Study II: feet and lower back—1 sensor per body part	Study I: 2-minute walk, 1-minute dual-task walk Study II: 7 days of unsupervised daily living monitoring	Study I: Accuracy: 85–88% Sensitivity: 80–89% Specificity: 87–88% Study II: less time spent freezing between people with and without FoG (*p* < 0.05)	Study I: OFF Study II: ON	Study I: CLT Study II: RL	Open-source threshold-based: freezing ratio (threshold)
Mazzetta et al. [25], 2019	7 PD-FoG+ Age range: 65–79 Mean H&Y: 2–3	Accelerometer, gyroscope and surface EMG Commercial IMUs	Shins and shanks—1 sensor per leg	TUG with obstacles, turning, door crossing	FOG detection: 2% false negatives 5% false positives	ON/ OFF	CLT, RL, VV	Gyro and sEMG fusion; custom real-time FOG index (threshold)
Caballol et al. [35], 2023	39 PD Mean age: 69 FoG status: NR Mean H&Y: NR	Accelerometer Commercial STAT-ON^TM^	Waist (left)—1 sensor	12-hour/day wear for 7 days; normal ADLs	Detected: FoG (23%) Kappa for FoG = 0.481	ON/ OFF	RL	ML: support vector machine
Demrozi et al. [36], 2020	10 PD (8 PD-FoG+; 2 PD-FoG-) patients from dataset; mean age: 66.5 Mean H&Y: 2.7 Dataset used: DAPHNET	Accelerometer Custom IMUs	Lower back, waist, ankle	Gait tasks with FoG, no-FoG and pre-FoG segments; labelled via video annotation	Pre-FoG detection: Sensitivity: 94.1% Specificity: 97.1% device latency: ~100–120 ms	OFF	Sim. RL	ML: k-NN classifier with PCA, LDA, kPCA, kLDA
Park et al. [22], 2024	14 PD-FoG+ patients Mean age: NR Mean H&Y: NR	Foot pressure sensors Commercial Pedar system	Both feet—multiple sensors per foot	Standardized 140 m walking path with dual-tasking, narrow corridors, turning, tray carrying	TCNN: Accuracy: 0.99 Precision: 0.68, sensitivity: 0.88, specificity: 0.99 F1-score: 0.76	ON	CLT, RL, VV	ML: temporal convolutional neural network
Tzallas et al. [37], 2014	Short term: 24 PD Long term: 20 PD Mean age: NR FoG status: NR Mean H&Y: NR	Accelerometer and gyroscope Commercial PERFORM IMUs	Wrists, ankles, waist—1 sensor per body part	Bed-to-chair walking, door opening, drinking; free-living for 5 days (~4 h/day)	FoG detection: 79% accuracy (short-term)	Mixed ON/ OFF	RL	ML: random forest
Tripoliti et al. [38], 2013	11PD-FoG+ 5 HC Mean age: 63 Mean H&Y: NR	Accelerometer and gyroscope Commercial ANCO IMUs	Accelerometers: both legs, both wrists, chest, waist—6 sensors; Gyroscopes: chest, waist—2 sensors	Standardized motor protocol: rising, walking, door crossing, water drinking	Random Forest: 96.11% accuracy Sensitivity: 81.94% Specificity: 98.74%	ON/OFF	Semi RL, VV	ML. naive Bayes, random forests, decision tree, random tree
Diep et al. [39], 2021	10 PD-FoG+ patients Mean age: 62.5 years Mean H&Y: NR	Accelerometer and gyroscope Commercial Mobility Lab Opal^TM^	Lateral shanks—1 sensor per body part	Stepping-in-place task for 100 seconds	General logistic model: AUC 0.81 Accuracy: 0.84, sensitivity: 0.86, specificity: 0.81	OFF	CLT	ML: binomial logistic regression
Reches et al. [23], 2020	71 PD-FoG+; mean age: 69.9 Used dataset: multicentric Mean H&Y: NR	Accelerometer, gyroscope and magnetometer Commercial Mobility Lab Opal^TM^	Lower back and both ankles—3 sensors per body part	FOG-provoking test in lab under 3 difficulty levels (single, dual motor, dual motor–cognitive)	SVM with RBF kernel: 86.6% accuracy, sensitivity: 80%, specificity: 82.5%	ON/ OFF	CLT	ML: support vector machine
Ashfaque et al. [40], 2021	10 PD-FoG+ Mean age: 66.4 Mean H&Y: 2.6 Dataset: DAHPNET	Accelerometer Commercial Mobility Lab Opal^TM^	Ankle, thigh, lower back—3 sensors per body part	Daily activities; annotated FOG and PreFOG (237 FOG events)	Best ensemble (M9): Accuracy: ~98.5% Precision: ~98%, sensitivity: ~98.5%, specificity: ~97.9%	NR	Sim. RL	ML: CNN, BiLSTM, ensemble models (machine learning)
Al-Adhaileh et al. [41], 2025	Multiple datasets: tDCS FoG (50 PD-FoG+), DeFOG (60 PD-FoG+), daily living (65 total incl. 45 PD-FoG+, 20 negative controls), Hantao’s (30 PD-FoG+); mean age: NR Mean H&Y: NR	Accelerometers, gyroscope, magnetometers, EMG and EEG (varies by dataset) Commercial IMUs, varies by dataset	Lower limbs (shins, thighs, ankles), waist; EMG on lower limb muscles, EEG on scalp Location and number of sensor placements varies by dataset	Controlled lab (FoG-provoking), home walking, week-long daily life monitoring	HTSAN model: AUC: 0.88–0.96 F1-score: 0.84–0.94 Accuracy: 85–98%	Mixed ON/ OFF	CLT, RL	ML: HTSAN
Bächlin et al. [42], 2010	10 PD-FoG+; mean age: 66.4 years Mean H&Y: 2.6	Accelerometer Custom IMUs	Shank, thigh, waist—3 sensor per body part	Straight walking, 360° turns, ADL simulations (e.g., fetching water)	Online detection: 73.1% sensitivity; 81.6% specificity	8 pts. OFF, 2 ON	CLT and RL	Proprietary FOG detection algorithm
Jovanov et al. [43], 2009	1 PD FoG status: NR 4 “simulated” PD-FoG+ Mean age: NR Mean H&Y: NR	Accelerometer and gyroscope Commercial Bosch SMB380	Right knee—1 sensor	Simulated FOG paths with sit-to-stand transitions and walking	Average detection latency of 332 ms, max latency of 580 ms; 0 false positives in 5 trials	NR	CLT	Rule-based algorithm with FFT (threshold)
Naghavi et al. [44], 2019	18 PD Mean age: 70.0 FoG status: NR Mean H&Y: NR	Accelerometer Commercial Mobility Lab Opal^TM^	Right and left ankles—1 sensor per body part	Obstacle-triggering path with narrow corridors, turns, stops	Best model ensemble): 97.4% FoG detection, 66.7% prediction, F1-score of 90.7% Sensitivity: 90.8%, specificity: 95%	NR	CLT	ML: ensemble: support vector machines, k-nearest neighbours, multi-layer perceptron
Shi et al. [45], 2020	67 PD-FoG+ Mean age: 69 Mean H&Y: NR	Accelerometer and gyroscope Custom IMUs	Both ankles and C7 vertebra—1 sensor per body part	7m TUG and simulated real-life setting	2D CNN (best ensemble) Accuracy: 89.2%, sensitivity: 82.1, specificity: 96%	NR	CLT, VV	ML: 2D CNN
May et al. [24], 2023	19 PD-FoG+ Mean age: 71.95 Mean H&Y: 2.7	Accelerometer and gyroscope Commercial Physilog and ActiGraph	Both feet and left side of waist—1 sensor per body part	Laboratory FoG tasks, simulated IADL tasks, 3-day unsupervised home monitoring	Mean detection accuracy > 90% (IADL tasks); strong correlation with video review (ρ = 0.77); home and lab sensor data correlation (ρ = 0.72)	ON	CLT and RL	ML: 2D CNN with continuous wavelet transform
Seuthe et al. [46], 2024	50 PD (22 PD-FoG+ 28 PD-FoG) Mean age: NR Mean H&Y: 2.1	Accelerometer Commercial Mobility Lab Opal^TM^	Gait initiation test: feet and lower back—1 sensor per body part For FoG detection: both shanks—1 sensor per body part	Gait initiation, overground walking, turning in place; single-task and dual-task conditions	Neither ML APA size nor APAPA size was significantly correlated with any FOG-related outcomes	ON	Sim. RL, RL	Modified pFOG algorithm, threshold: pFOG > 0.7
Belluscio et al. [47], 2019	32 PD (15PD-FoG+ 17 PD- FoG- and 8 HC Mean age: 67 Mean H&Y: NR	Accelerometer, gyroscope, magnetometer and fNRIS Commercial Mobility Lab Opal^TM^	IMUs: sternum, pelvis, wrists, shanks, both feet—1 sensor per body part fNRIS—forehead	360° turning-in-place for 2 minutes under single-task and dual-task conditions	Higher PFC activity is correlated with worse FOG in PD-FOG+ patients (p.0.048) and smaller number of turns in PD-FOG+ (P 0.02)	OFF	CLT	Signal preprocessing of fNIRS (HbO_2_/HHb). IMU-derived FoG ratio (threshold)
Goris et al. [14], 2025	177 PD (54 PD-FoG-, 22 PD-FoG+ (aware), 82 PD-FoG+ (unaware) Mean age: 62.56 Mean H&Y: 1–3	Accelerometer and gyroscope Commercial Mobility Lab Opal^TM^	Both shins, lower back—1 sensor per body part Feet-mounted sensors were excluded from final analysis	1-minute 360° alternating turn with cognitive dual-task	Best AUC = 0.65, sensitivity: 68.3%, specificity: 61.7%	ON	CLT	ML: FOG index derived from FOG ratio; frequency domain analysis; ROC analysis (threshold)
Arami et al. [26], 2019	10 PD-FoG+ Mean age: 66.5 Mean H&Y: 2.6 Dataset: DAPHNET	Accelerometer Custom IMUs	Lower back, shank, thigh—3 sensors	Walking trials with turns and simulated ADLs	Sensitivity: 93% Specificity: 91%	ON	CLT, RL	ML: SVM and PNN
Hu et al. [48], 2023	21 PD FoG status: NR Mean age: NR Mean H&Y: NR	Foot pressure sensor—1 sensor Commercial Zeno Walkway	Feet—1 sensor (walkway)	TUG trials with FoG provoking elements	Sensitivity: 83.4% Specificity: 72.9% Accuracy: 75.7% AUC: 0.85	O/OFF	CLT	ML: adversarial spatial–temporal network
Mazilu et al. [49], 2015	18 PD-FoG+ Mean age: 68.9 Mean H&Y: 2-4 Dataset: CuPiD	ECG and SC Commercial Shimmer sensors	Sternum and wrist—2 sensors	Walking trials with FoG provoking elements	71.3% of FoG episodes predicted with SC sensor	ON	CLT	Anomaly-based algorithm (threshold)
Mikos et al. [50], 2019	63 PD-FoG+ Mean age: 68.9 Mean H&Y: 2.5	Accelerometer, gyroscope and magnetometer Commercial	Ankle—1 sensor	Walking trials with turns and narrow spaces	Sensitivity: 95.6% Specificity: 90.2%	NR	CLT	ML: neural network
Murtaza et al. [51], 2025	12 PD-FoG+ patients Mean age: 69.1 Mean H&Y: not reported	Accelerometers, gyroscopes, EEG, EMG and SC Commercial IMUs	Waist, both shanks (IMUs), left wrist finger (SC), mastoid process (EEG), shins (EMG)	Walking trials with turns, stops, avoiding obstacles	EMG and IMUs data (best combination) F1 score: 98.82%	OFF	CLT	ML: SVM
Noor et al. [52], 2021	10 PD-FoG+ Mean age: NR Mean H&Y: NR Dataset: DAPHNET	Accelerometer Custom IMUs	Shank, thigh, lower back—3 sensors	Three walking trials with ADLs simulation	Sensitivity: 90.94% Specificity: 67.04%	NR	CLT	ML: naïve Bayes, SVM with RBF kernel, SVM with polynomial kernel, random forest, ensemble voting
Pierleoni et al. [53], 2019	10 PD-FoG+ Mean age: 67-7 Mean H&Y: NR	Accelerometer, gyroscope and magnetometer Commercial	Feet—1 sensor on each foot	TUG, walking through narrow spaces	99.7% accuracy for FoG detection	ON	CLT	Freeze index (threshold)
Prado et al. [54], 2020	8 PD-FoG+ and 2 FoG- Mean age: 67.9 Mean H&Y: 2.8	accelerometer, gyroscope, foot pressure sensors Commercial DeepSole system	Feet—12 sensors per foot	7-meter Zeno Walkway	Sensitivity: 96.0% Specificity: 99.6% Accuracy: 99.5%	NR	CLT	ML: CNN
Shi et al. [55], 2022	63 PD-FoG+ Mean age: 69.4 Mean H&Y: NR	Accelerometer, gyroscope and magnetometer Custom IMUs	Ankle—1 sensor on each ankle	TUG and second walking trial with FoG provoking elements	F1 score: −91.5%	NR	CLT	ML: CNN
Tahafchi et al. [56], 2019	4-PD-FoG+ Mean age: NR Mean: H&Y: NR	Accelerometer, gyroscope and EMG Commercial Shimmer IMUs	Feet, shanks—2 sensors on each side	Walking trials designed to trigger FoG	AUC: 0.906-0.963	NR	CLT	ML: CNN

2MWT = 2-minute walk test, 5Fold-CV = 5-fold cross-validation, ADL = activities of daily living, AUC = area under the curve, APA = anticipatory postural adjustment, AP = anteroposterior, AUROC = area under the receiver operating characteristic curve, BiLSTM = bidirectional long short-term memory, CLT = controlled laboratory testing, CNN = convolutional neural network, COP = centre of pressure, CuPiD = Clinical Decision Support System and Patient Interaction Platform for Parkinson’s Disease, CWT = continuous wavelet transform, ECG = electrocardiography, EEG = electroencephalography, EMG = electromyography, FFT = fast Fourier transform, fNRIS = functional near-infrared spectroscopy, FoG = freezing of gait, FNR = false negative rate, FPR = false positive rate, HbO2 = oxyhaemoglobin, HC = healthy control, HHb = deoxyhaemoglobin, HTSAN = hierarchical temporal spatiotemporal attention network, H&Y = Hoehn–Yahr, IADL = instrumental activities of daily living, IMU = inertial measurement unit, k-LDA = kernel linear discriminant analysis, k-NN = kernel nearest neighbours, k-PCA = kernel principal component analysis, LDA = linear discriminant analysis, LOSO = leave one subject out, ML = machine learning, MLa = mediolateral, NN = neural network, NR = not reported, OFF = off medication state, ON = on medication state, PCA = principal component analysis, PD = Parkinson’s disease, PFC = prefrontal cortex, PNN = probabilistic neural network, PP = plantar pressure, PS = past samples, RL = real life, RBF = radial basis function, RNN = recurrent neural network, ROC = receiver operating characteristic, RQA = recurrence quantification analysis, RUS = random undersampling, SC = skin conductance, SVM = support vector machine, SLT = sagittal leg tilt, sEMG = surface electromyography, TCNN = temporal convolutional neural network, TUG = timed up and go test, VV = video validation.

**Table 3 sensors-25-05101-t003:** Number and ratio of studies using different sensors or combinations of sensors.

Sensor Type for FoG Detection	Number of Studies	Percentage of Total Number of Studies
Accelerometer	10	23.2%
Accelerometer + gyroscope	11	25.5%
Accelerometer + gyroscope + magnetometer	9	20.9%
Accelerometer + plantar pressure sensor	1	2.3%
Accelerometer + gyroscope + plantar pressure sensor	2	4.7%
Accelerometer + gyroscope + force sensing resistors	1	2.3%
Accelerometer + gyroscope +muscle activity sensors	1	2.3%
Accelerometer + gyroscope + sEMG	2	4.7%
Accelerometer + gyroscope + magnetometer + sEMG	1	2.3%
Accelerometer + gyroscope + EMG + SC sensor	1	2.3%
Plantar pressure sensor	3	7.0%
ECG + SC sensor	1	2.3%

ECG = electrocardiography, EMG = electromyography, FoG = freezing of gait, SC = skin conductance, sEMG = surface electromyography.

**Table 4 sensors-25-05101-t004:** Studies reporting the best FoG detection performance metrics for each sensor combination.

Sensor Type for FoG Detection	No. of Articles	Sensitivity (%)	Specificity (%)	Accuracy (%)	Other Performance Metrics	Algorithm	Author of Article with Best Performance
Accelerometer	10	98.5	97.9	98.5	-	ML	Ashfaque et al. [40]
Accelerometer + gyroscope	11	~100	~100	-	Avg. error rate: < 5%	ML	Chomiak et al. [21]
Accelerometer + gyroscope + magnetometer	9	83	98	96	-	M/ Threshold	Antonini et al. [30]
Accelerometer + plantar pressure sensor	1	96	94	-	-	Threshold	Marcante et al. [18]
Accelerometer + gyroscope + plantar pressure sensor	2	96.0	99.6	99.5	-	ML	Prado et al. [54]
Accelerometer + gyroscope + force sensing resistors	1	78.4	91.7	88.1	F1 score: 0.78	ML	Ren et al. [28] *****
Accelerometer + gyroscope + muscle activity sensors	1	2.7	97.2	95.0	F1 score: 0.023	ML	Slemenšek et al. [20]
Accelerometer + gyroscope + magnetometer + sEMG	1	-	-	85-97	HTSAN model: AUC 0.88–0.96 F1-score: 0.84–0.94	ML	Al-Adhaileh et al. [41]
Accelerometer + gyroscope + sEMG	2	-	-	-	FoG detection: 2% false negatives and 5% false positives	Threshold	Mazzetta et al. [25]
Accelerometer + gyroscope + EMG + SC sensor	1	-	-	-	F1 score: 0.99 (EMG and IMUs data)	ML	Murtaza et al. [51]
Plantar pressure sensor	3	88	99	99	F1 score: 0.76	ML	Park et al. [22]
ECG + SC	1	-	-	-	71.3% of FoG episodes predicted with SC sensor	Threshold	Mazilu et al. [49]

AUC = area under the curve, ECG = electrocardiography, EMG = electromyography, FoG = freezing of gait, HTSAN = hierarchical temporal spatiotemporal attention network, ML = machine learning, SC = skin conductance, sEMG = surface electromyography. * The initial feature extraction in this study consisted of a combination of an accelerometer, gyroscope and force sensing resistor; however, after consideration of multiple factors such as detection performance and deployment cost, the final and best feature extraction only used an accelerometer and gyroscope at the left shank, achieving the performance metrics presented in Table 4.

**Table 5 sensors-25-05101-t005:** Best FoG detection performance metrics and algorithms for each sensor type or sensor combination for FoG detection in a real-life setting. AUC= area under the curve, FoG = freezing of gait, IADL = instrumental activities of daily living, ML= machine learning.

Sensor Type for FoG Detection	No. of Articles	Sensitivity (%)	Specificity (%)	Accuracy (%)	Other Performance Metrics	Sensor Integration Method and Placement	Algorithm	Author of Article with Best Performance
Accelerometer	5	87.7	88.3	-	50% predicted FoG 3.1s before onset, 50% detected with 0.8 s delay	STAT-ON^TM^—single device worn on waist	ML	Borzi et al. [32]
Accelerometer + gyroscope + magnetometer	1	-	-	-	Time spent freezing differentiated FoG+ and FoG-	The Opal V2R^TM^—1 sensor on each foot and 1 sensor on lower back	Threshold	Mancini et al. (study II) [34]
Accelerometer + gyroscope	2	-	-	>90%	Strong correlation with video review (ρ = 0.77) Home and laboratory sensor data correlation (ρ = 0.72)	1 sensor on each foot and 1 sensor on waist	ML	May et al. [24]
Accelerometer + gyroscope + magnetometer (DeFoG dataset) Accelerometer + gyroscope (Daily Living dataset)	1	-	-	DeFOG: 87% Daily living: 85%	DeFOG: AUC: 0.91 F1 score: 0.88 Daily: AUC: 0.88 F1 score: 0.84	DeFOG dataset: 1 sensor on each ankle Daily living: 1 sensor on each ankle and 1 sensor on waist	ML	Al-Adhaileh et al. [41]

## Data Availability

No new data were created or analyzed in this study. Data sharing is not applicable to this article.

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
