# Peer review of "The Usefulness of Wearable Sensors for Detecting Freezing of Gait in Parkinson’s Disease: A Systematic Review"

_sensors, 2025, doi:10.3390/s25165101_

Round 1

Reviewer 1 Report

Comments and Suggestions for Authors

Summary

The systematic review offers insights into the use of wearables for FoG detection. It included valuable information from recent studies, but from a limited source of search.

Minors

  1. Please list the reasons for the excluded articles from the 101 results in PubMed. Figure 1 is not clear why 56 studies are excluded, and the other reasons for excluding 11 studies are not specified.
  2. I think it is important to introduce the behavioral manifestations of FoG in the introduction and what trigger events are. If this information can be extracted from the reviewed studies, it will add more valuable information for the systematic review.
  3. How FoG is measured should be introduced. What is the ground truth of FoG for the studies reviewed?
  4. The severity of PD patients is missing in the systematic review, which is important information to report.
  5. It may be helpful to add what models of sensors (commercial or customized?) are used in the reviewed studies.
  6. It would be nice to provide some insights into how wearable sensors can predict, measure and quantify FoG, in addition to detection.
  7. The single source search on PubMed limited the scope of searching. It does not include all recent studies of using wearables to detect FoG. Maybe consider adding IEEE and ACM into the source for searching.

Author Response

Reviewer 1

Summary

The systematic review offers insights into the use of wearables for FoG detection. It included valuable information from recent studies, but from a limited source of search.

Minors

1. Please list the reasons for the excluded articles from the 101 results in PubMed. Figure 1 is not clear why 56 studies are excluded, and the other reasons for excluding 11 studies are not specified.

RE: Figure 1 is now updated with more precise explanation of the reasons for exclusion of studies (page 4, lines 152-156)

2. I think it is important to introduce the behavioral manifestations of FoG in the introduction and what trigger events are. If this information can be extracted from the reviewed studies, it will add more valuable information for the systematic review.

RE: Behavioural manifestations and trigger events were added in lines 60-65. Additionally, some trigger events used to elicit FoG were cited from reviewed articles in lines 96-98.

3. How FoG is measured should be introduced. What is the ground truth of FoG for the studies reviewed?

RE: The way FoG is measured is described in lines 99-107.

4. The severity of PD patients is missing in the systematic review, which is important information to report.

RE: Added Hoehn-Yahr stages for each article that reported it in Table 2. Additionally, Hoehn-Yahr range and average are described in the results section in lines 186-187.

5. It may be helpful to add what models of sensors (commercial or customized?) are used in the reviewed studies.

RE: Added sensor models (custom/commercial) for each article in Table 2. We also described the ratio between commercial and custom IMUs use and the most used brand of commercial IMUs in lines 221-223.

6. It would be nice to provide some insights into how wearable sensors can predict, measure and quantify FoG, in addition to detection.

RE: FoG measurement, quantification and prediction are described in lines 99-107 in the Introduction section. In addition, the capability of the sensors to predict and quantify FoG is given in Table 2 (Column “Main Results”) and discussed throughout the paper.

7. The single source search on PubMed limited the scope of searching. It does not include all recent studies of using wearables to detect FoG. Maybe consider adding IEEE and ACM into the source for searching.

RE: We expanded the search to IEEE and ACM, which tripled the total article sum, and added 10 articles in the review that met inclusion criteria. The rest were duplicates or articles that did not meet the inclusion criteria. This is now reflected in the updated Methodology section, including updated Figure 1 and updated Table 1 in the manuscript.

Reviewer 2 Report

Comments and Suggestions for Authors

Dear Editor,

thank you for sending the article titled: Usefulness of wearable sensors for detecting freezing of gait in Parkinson’s disease: a systematic review for review process. Article seems interesting at firs sight, but it should be corrected as follows:

  • line 145-153 Please present statistical results on the graphs (Mean (SD))
  • It will be great if authors include in the Introduction section 1-2 paragraphs about IMU's, which can be used in this kind of experiment. For example in the article titled: Human gait feature detection using inertial sensors wavelets, Biosystem and Biorobotics Proceedings of the 2nd International Symposium on Wearable Robotics, WeRob 2016, Springer 2017, 397-402 authors used IMU for obtain kinematic parameters. To improve article quality I suggest cite above manuscript and write 1-2 paragraphs in the Introduction section
  • Table 2 is non-readable - please shorten text inside table. Maybe it will be better present  some parameters as figure?
  • Figure 2 - Please place a sketch of the human and the location of the IMU on it
  • Figure 3 - as above
  • lack of limitation of the study

Author Response

Reviewer 2

Dear Editor,

thank you for sending the article titled: Usefulness of wearable sensors for detecting freezing of gait in Parkinson’s disease: a systematic review for review process. Article seems interesting at firs sight, but it should be corrected as follows:

  • line 145-153 Please present statistical results on the graphs (Mean (SD))

RE: We expanded the statistical information on sample size, age distribution and H&Y scale by adding median and IQR in addition to the range (min-max) and mean±standard deviation (lines 181-186) and added Figure 2 for the patients’ FoG status distribution (line 200).

  • It will be great if authors include in the Introduction section 1-2 paragraphs about IMU's, which can be used in this kind of experiment. For example in the article titled: Human gait feature detection using inertial sensors wavelets, Biosystem and Biorobotics Proceedings of the 2nd International Symposium on Wearable Robotics, WeRob 2016, Springer 2017, 397-402 authors used IMU for obtain kinematic parameters. To improve article quality I suggest cite above manuscript and write 1-2 paragraphs in the Introduction section

RE: Paragraph on IMUs from the suggested source was added in lines 86 – 98.

  • Table 2 is non-readable - please shorten text inside table. Maybe it will be better present some parameters as figure?

RE: Table 2 has now been optimised by shortening the text and including only the necessary information in it.

  • Figure 2 - Please place a sketch of the human and the location of the IMU on it

RE: We added a drawing of the human body with IMUs location. Now marked as Figure 3 between lines 257-262.

  • Figure 3 - as above

RE: We added a drawing of the human body with IMUs location. Now marked as Figure 4 between lines 321-326.

  • lack of limitation of the study

RE: A section on the imitations of the study is now added in the Discussion section – lines 496-507.

Reviewer 3 Report

Comments and Suggestions for Authors

This manuscript provides a review of the application and value of wearable sensors in detecting freezing of gait (FoG) in Parkinson’s disease, offering valuable insights for the field. To further enhance the quality of the manuscript, I would like to offer the following suggestions:

(1) The abstract is somewhat lengthy for an academic paper. It is recommended to shorten it appropriately to improve clarity and conciseness.

(2) For the tables, please follow the standard three-line table format. Additionally, it is not advisable to allow tables to span across pages. If this is unavoidable, please ensure that the appropriate multi-page formatting is applied.

(3) In Table 3, the term “ratio” is unclear. What does it represent? Which two values are being compared?

(4) Figures 2 and 3 are difficult to interpret. Please clarify what the x- and y-axes represent, and consider labeling them directly in the figures for better readability.

(5) In the Discussion section, it would be helpful to outline the current limitations or challenges of this technology, along with potential strategies to address them.

(6) For a review article, the number of references (just over 50) seems relatively low. It is recommended to include more relevant and recent literature to strengthen the comprehensiveness of the review.

Author Response

Reviewer 3

This manuscript provides a review of the application and value of wearable sensors in detecting freezing of gait (FoG) in Parkinson’s disease, offering valuable insights for the field. To further enhance the quality of the manuscript, I would like to offer the following suggestions:

(1) The abstract is somewhat lengthy for an academic paper. It is recommended to shorten it appropriately to improve clarity and conciseness.

RE: Abstract is now shortened to 387 words.

(2) For the tables, please follow the standard three-line table format. Additionally, it is not advisable to allow tables to span across pages. If this is unavoidable, please ensure that the appropriate multi-page formatting is applied.

RE: A three-line table format has been used. Table 2 is now optimised regarding the quantity of text it contains, but it still spans across pages. Further shortening of the table would lead to a loss of important information relevant to the study.

(3) In Table 3, the term “ratio” is unclear. What does it represent? Which two values are being compared?

RE: The term “ratio” was changed to percentage of total number studies in Table 3 to improve clarity.

(4) Figures 2 and 3 are difficult to interpret. Please clarify what the x- and y-axes represent, and consider labeling them directly in the figures for better readability.

RE: Figures 2 and 3, now Figures 2 and 4 respectively are completely changes and depict a human body with IMUs locations and number of articles on it.

(5) In the Discussion section, it would be helpful to outline the current limitations or challenges of this technology, along with potential strategies to address them.

RE: This has now been done appropriately throughout the Discussion section, for example see lines 391-396, 408-409, 422-424, 433-435, 451-455, 465-469, etc.

(6) For a review article, the number of references (just over 50) seems relatively low. It is recommended to include more relevant and recent literature to strengthen the comprehensiveness of the review.

RE: Expanded the number of cited sources to 69 by expanding the search to IEEE and ACM articles.

Round 2

Reviewer 3 Report

Comments and Suggestions for Authors

This manuscript has been improved after revision.